# Synthesis and Thermal Degradation Study of Polyhedral Oligomeric Silsesquioxane (POSS) Modified Phenolic Resin

**DOI:** 10.3390/polym13081182

**Published:** 2021-04-07

**Authors:** Degang Wang, Jie Ding, Bing Wang, Yingluo Zhuang, Zhixiong Huang

**Affiliations:** 1College of Aerospace Science and Engineering, National University of Defense Technology, 109 Deya Road, Changsha 410073, China; darwin_wang@163.com or; 2School of Materials Science and Engineering, Wuhan University of Technology, 122 Luoshi Road, Wuhan 430070, China; wangbingwhut@whut.edu.cn (B.W.); zhuangyingluo@whut.edu.cn (Y.Z.); zhixiongh@whut.edu.cn (Z.H.)

**Keywords:** polyhedral oligomeric silsesquioxane, phenolic resin, thermal degradation, residual char yield

## Abstract

In this paper, a new polyhedral oligomeric silsesquioxane containing a phenol group (POSS-Phenol) is prepared through the Michael addition reaction, which is added to the synthesis of phenolic resin as a functional monomer. Infrared spectroscopy (IR) is used to demonstrate the chemistry structure of the synthesized POSS modified phenolic resin. After introducing POSS into the resole, a comprehensive study is conducted to reveal the effects of POSS on the thermal degradation of phenolic resin. First, thermal degradation behaviors of neat phenolic resin and modified phenolic resin are carried out by thermogravimetric analysis (TGA). Then, the gas volatiles from thermal degradation are investigated by thermogravimetric mass spectrometry (TG-MS). Finally, the residues after thermal degradation are characterized by X-ray diffraction (XRD). The research indicates that POSS modified phenolic resin shows a better thermal stability than neat phenolic resin, especially at high temperatures under air atmosphere. On the one hand, the introduction of the POSS group can effectively improve the release temperature of oxygen containing volatiles. On the other hand, the POSS group forms silica at high temperatures under air, which can effectively inhibit the thermal oxidation of phenolic resin and make phenolic resin show a better high-temperature oxidation resistance.

## 1. Introduction

Phenolic resin is an indispensable foundational material in the field of aerospace thermal protection materials due to its excellent thermal stability, solvent resistance and structural integrity, and it is also a precursor of C/C composite materials [1,2]. However, some of the properties of phenolic resin still need to be further improved to meet the growing performance requirements of aerospace thermal protection materials [3,4,5]. The adverse characteristics that need to be ameliorated include a moderate thermal oxidation stability, inherent brittleness and residual char yield [6,7]. For example, the residual char yield of phenolic resin is mainly over 65% after being pyrolyzed under atmosphere without oxygen at 800 °C. However, the residual char yield of phenolic resin is only about 7% after being pyrolyzed under air at the same temperature.

In recent years, nanotechnology has been widely used in polymer modification. The volume effect, surface effect and quantum size of nanomaterials have made significant breakthroughs in modified phenolic resins, such as increasing the heat resistance, ablation resistance and thermal insulation [8,9]. Polyhedral oligomeric silsesquioxane (POSS) is a new type of silicone inorganic–organic material with a cubic cage-shaped nanostructure and a multiplicity of functional groups, which is gradually attracting attention in polymer modification. Unlike traditional organic materials, POSS, as a silicone compound, has the advantages of being nonvolatile, odorless, environmentally friendly, and so on. According to current reports, POSS modified polymers include polyethylene [10], vinyl ester [11,12,13], epoxy resin [14,15], phenolic resin [16,17,18], etc.

Introducing POSS into the cross-link network structure of phenolic resin by means of a chemical bond can avoid the universal reuniting of nanoparticles, which can improve the glass transition temperature and residual char yield of phenolic resin. In particular, in extremely high temperature conditions, the SiO_2_ layers formed after the degradation of POSS have a good heat oxygen effect, which will greatly improve the thermal oxidation resistance of phenolic resin [19,20].

At present, the research work on phenolic resin mainly focuses on chemical modification and the synthesis process, while the thermal degradation study of modified phenolic resin has been less reported. As an important thermal protective ablation material substrate, the degradation properties of modified phenolic are closely related to its application under a high-temperature ablation environment [21]. On the other hand, the degradation study of the modified phenolic resin can provide rational feedback for the modification of the resin. Regarding the thermal stability of phenolic resin, the three-stage degradation mechanism proposed by Trick is the most representative for this [22]. The thermal degradation of phenolic resins is divided into three stages as the temperature increases. The first stage (–300 °C) is a further condensation reaction between phenolic hydroxyl groups or methylene (–CH_2_–) groups; In the second stage (300–800 °C), the methylene and ether bonds are broken to produce main products such as methane, hydrogen and carbon monoxide; In the third stage (800 °C–), hydrogen atoms are separated from the carbon skeleton to generate hydrogen and a continuous network of glassy carbon structure.

In this article, we used the hydrolysis and condensation of trimethoxysilane to prepare POSS-8SH. Then, the POSS-8Phenol, which contains octagonal phenol, was synthesized through a mild and efficient Michael addition reaction between sulfhydryl and double-bonded C=C. After this, POSS-8Phenol, as a special phenol, was introduced into the molecular structure of thermoset phenolic resin. The chemical structure of POSS modified phenolic resin was studied by IR. The thermal degradation behaviors of POSS modified phenolic resin were respectively studied by TGA under argon and air. Moreover, the volatiles and residues of POSS modified phenolic resin were characterized by TG-MS and XRD in order to further understand the degradation mechanism. Neat phenolic resin was fabricated into a subsample and investigated with the same analysis process.

## 2. Materials and Methods

### 2.1. Materials

(3-mercaptopropyl) trimethoxysilane (MPT), dibutylamine was purchased from Aladdin Reagent Co., Ltd. (Shanghai, China). Phenol, 37% aqueous solution of formaldehyde, sodium hydroxide, methanol, acetone, dichloromethane, hydrochloric acid, anhydrous magnesium sulfate and sodium chloride were purchased from Sinopharm Chemical Reagent Co., Ltd. (Wuhan, Hubei, China). N-(4-hydroxyphenyl) maleimide (HPM) was purchased from Weng Jiang Reagent Co., Ltd. (Shaoguan, Guangdong, China). All the chemicals were used directly without further purification.

### 2.2. Sample Preparation

Synthesis of POSS-8SH: POSS-8SH is prepared via a dehydration condensation reaction of trimethoxysilane, according to the literature methods [11,23]. Briefly, 15.00 mL MPT was dissolved in 360.00 mL methanol solution, and then 20.00 mL of concentrated hydrochloric acid was dripped for 30 min. The mixture was stirred and refluxed at 90 °C for 24 h. A viscous white liquid could be obtained at the bottom of the solution after standing. The supernatant liquid was decanted, and the precipitate was washed several times with methanol to remove unreacted MPT and other impurities. Finally, POSS-8SH was obtained after CH_2_Cl_2_ was evaporated under vacuum.

Synthesis of POSS-8Phenol: The synthesis of POSS-8Phenol occurred through the Michael addition reaction between the sulfhydryl group and the electron-deficient double bond of the maleimide group under the catalysis of an organic base [24,25,26]. 3.00 g POSS-8SH was dissolved in acetone, and 0.30 g (1 mol% of HPM) N-dibutylamine was added as a catalyst. After that, 4.41 g HPM was dropped for 30 min, in which the molar ratio of the mercapto group to maleimide group was 1:1. The mixed system was stirred at 25 °C for 24 h. After the reaction, the mixed solution was crystallized from deionized water/acetone as a light yellow crystalline solid, which was illustrated in Figure 1.

Preparation and curing of POSS modified phenolic resin: 50% NaOH aqueous solution was used to catalyze the synthesis of thermosetting phenolic resin, in which the molar amount of NaOH was 1% of phenol. 94.10 g (1 mol) phenol, 12.66 g (0.005 mol) POSS-8phenol, and 126.95 g of 37% formaldehyde aqueous solution (molar ratio of phenol group: formaldehyde is 1.5) were blended with a 250.00 mL flask equipped with stirring and reflux. The mixed solution was heated to 70 °C for 2 h, then heated to 95 °C for 20 min and dehydrated for 30 min after holding for 1 h. The obtained modified phenolic resin was about 130.00 g. The synthesis steps of the neat phenolic resin were consistent with the above process, except for POSS-8phenol. The obtained neat phenolic resin was about 150.00 g. Both phenolic resins were placed in the closed molds with an external pressure of 10.00 MPa, and the cured procedure was at 80 °C (4 h)/120 °C (2 h)/180 °C(4 h).

### 2.3. Sample Characterization

The chemical modification of phenolic resin by POSS was investigated by Fourier transform infrared spectroscopy, on KBr pellets from 400 cm^−1^ to 4000 cm^−1^ by a Nicolet Nexus IR Spectra, (Madison, WI, USA). The thermal stabilities of POSS modified phenolic were characterized by thermogravimetric analysis (TGA, NETZSCH STA449c/3/G) under argon and air. The TGA sample was about 5.00 mg and was heated at temperatures of 50~850 °C at a constant heating rate of 10 °C/min. The flow rate of argon was about 50.00 mL/min. The degradation volatiles of POSS modified phenolic resin were studied by thermogravimetric mass spectrometry (TG-MS, NETZSCH, 409-MS-Skimmer connected to the TGA system) under argon and air, and the heating process was the same as that of TGA. For the TG-MS measurements, about 4.00 mg of the sample was taken onto a platinum pan, and the scanned mass range was *m*/*z* = 1.6–100. After determining the target species, the final measurements were performed by setting the mass spectrometer at the selected ion monitoring (SIM) mode. The phase compositions of the POSS modified phenolic resin and its degradation residues were characterized by X-ray diffraction (XRD, D8 Advance, AXS, Germany). Every sample was scanned from a 2θ angle of 10–70 ° at a scan rate of 5 °/min. Under the same conditions, neat phenolic resin was also investigated by the above characterizations.

## 3. Results and Discussion

### 3.1. Synthesis of Modified Phenolic Resin

The IR spectra of neat phenolic resin and modified phenolic resin are shown in Figure 2 to confirm the chemical structure of POSS in modified phenolic. Compared with the IR spectrum of neat phenolic resin, the IR spectrum of modified phenolic resin has two more characteristic absorption peaks at 1703 cm^−1^ and 1096 cm^−1^. The absorption peak at 1703 cm^−1^is assigned to C=O in the maleimide group of POSS-8Phenol, and the characteristic of Si–O–Si stretching is at 1096 cm^−1^, these being consistent with the literature [27]. Other infrared absorption peaks of the modified resin are basically the same with those of the unmodified resin. The distributions of characteristic peaks are as follows: the characteristic peaks of O–H of the phenol group and –CH_2_OH are at around 3500 cm^−1^ (broad peak); the C–H stretching vibration absorption peaks in –CH_2_– range from 2800 cm^−1^ to 2950 cm^−1^; the stretching vibration peaks of the C=C double bond in the benzene ring appear at 1610 cm^−1^ and 1510 cm^−1^; the characteristic absorption peak of C–O in the phenolic group is at 1270 cm^−1^; the absorption peaks of the aliphatic C–O stretching vibration are at 1130 cm^−1^ and 1005 cm^−1^; and the ortho- and para-substituted peaks of the benzene ring are at 760 cm^−1^ and 870 cm^−1^. All these absorption peaks are also reported in the synthesis results of phenolic resin elsewhere [28,29]. It can be confirmed that POSS-8Phenol has been incorporated into the structure of phenolic resin.

### 3.2. Thermogravimetric Analysis

Thermogravimetric analysis is widely used to determine the thermal stability of polymer materials. The advantage is that the relationship between the mass loss percentage and the temperature can be observed from the obtained curve, and the thermal stability of the material can be more intuitively evaluated [30]. Figure 3 shows the TG/DTG curves of neat phenolic and modified phenolic under argon and air at a heating rate of 10 °C/min. Before 300 °C, as reported in Ref. [31], the slight weight loss is mainly the removal of water in phenolic resin. The thermal degradation behavior of the phenolic resin mainly occurs above 300 °C.

The thermogravimetric curves of neat phenolic resin under argon and air atmosphere are shown in Figure 3a,b. The weights under argon and air atmosphere show no obvious changes below 300 °C. The fastest weight loss is at about 547 °C under argon, and the residual weight of neat phenolic resin is 70.82% at 800 °C. By contrast, the temperature of the fastest weight loss is only about 494 °C under air, which is significantly lower than that of neat phenolic resin under argon. In addition to the weight loss peak at 494°C, there is another slight weight loss peak at 540 °C during the thermal degradation process of neat phenolic resin under air atmosphere. The weight of neat phenolic resin is almost unchanged above 600 °C, and its residual weight is only 6.95% at 800 °C.

The thermogravimetric curves of POSS modified phenolic resin under argon and air atmosphere are shown in Figure 3c,d. The temperature of the fastest weight loss is about 617°C under argon, which is almost 70 °C higher than that of neat phenolic resin. The residual weight of POSS modified phenolic resin is 77.80% at 800 °C, which increases by 6.98% in comparison with that of neat phenolic resin. In accordance with the thermal behaviors under argon, the thermal stability of POSS modified phenolic resin is better than that of neat phenolic resin under air atmosphere. The temperature of the fastest weight loss is about 515 °C under air, and another slight weight loss peak is at about 632 °C. The temperatures corresponding to these two pyrolysis peaks of POSS modified phenolic resin are 21 °C and 92 °C higher than those of phenolic resin, respectively. The residual weight of POSS modified phenolic resin is 21.68% at 800°C under air, which evidently increases by 14.73% in comparison with that of neat phenolic resin.

Many studies on the thermal degradation of phenolic resin under inert gas atmosphere could be found elsewhere [22,30,31,32]. The thermal degradation process of phenolic resin under inert gas atmosphere can be divided into three stages. In the first stage, it begins with the formation of additional intermolecular crosslinks between aromatic rings. In the second one, the weight loss mainly happens because of the instability of methylene groups. Finally, hydrogen results from the splitting of hydrogen atoms directly bonded to benzene nuclei. The weight loss mainly occurs in the second stage, and the breakage of methylene leads to the decomposition of the whole carbon chain structure of phenolic resin. As shown in Figure 3a,c, the introduction of POSS does not essentially change the degradation process of phenolic resin under argon. The degradation process of phenolic resin under argon is still divided into three stages. Although the thermal degradation temperature of the modified phenolic resin is increased by about 70 °C, the residual char yield of the modified phenolic resin is not significantly increased at 800 °C.

Different from the thermal degradation result under argon, the residual char yield of the modified phenolic resin is significantly increased with the elevation of the depth of thermal degradation under air. As shown in Figure 3b,d, the residual char yield of modified phenolic resin at 800 °C is 14.73% higher than that of phenolic resin at 800 °C. It is well known that phenolic resin is easier to crack at elevated temperatures under air than under argon [33]. Owing to the presence of oxygen, the methylene groups produced by thermal degradation are more vulnerable to be attacked [31,34]. With the increase of the temperature, the thermal oxidation effect becomes stronger. Oxygen in the air can directly oxidize the intermediate products of phenolic resin to carbon monoxide or carbon dioxide. The introduction of POSS can inhibit the thermal oxidation effect, so that phenolic resin has a better oxidation resistance at elevated temperatures.

### 3.3. Degradation Volatiles’ Analysis

In order to understand the thermal degradation mechanism of phenolic resin, the gas produced by pyrolysis is further investigated under argon and air atmosphere at elevated temperatures. Mass spectrometry is used to detect small molecular volatiles during the thermal degradation process of neat phenolic resin and POSS modified phenolic resin. The results of the TG-MS study under argon and air are presented in Figure 4. As previously reported [31,35], the species with *m*/*z* of 2, 16, 18, 28, 30, 44 and 94 are assigned to hydrogen, methane, water, carbon monoxide, ethane, carbon dioxide and phenol, respectively.

As shown in Figure 4a,c, under argon, most small molecules produced by the degradation of the neat phenolic reach the maximum peaks at 580–610 °C; such a phenomenon is similar to other experimental results reported elsewhere [32], while the POSS modified phenolic shows the maximum peaks of gas emission at about 630–650 °C. As shown in Figure 4b,d, the corresponding gas evolutions under air are more complex than those under argon. Under air, the neat phenolic shows the maximum peaks of gas emission at about 480–520 °C, consistent with previous research results [32], whereas volatiles such as hydrogen, methane, water, carbon dioxide and phenol produced by the thermal degradation of POSS modified phenolic resin reach the maximum peaks at 540–610 °C. TG-MS study shows that the POSS modified phenolic resin represents better thermal stability than neat phenolic resin under air atmosphere, which is consistent with the result of TGA.

The main chain of phenolic resin is connected by methylene (–CH_2_–) groups. Trick et al. pointed out that the methylene groups are easily attacked and broken during the thermal degradation process of phenolic resin [22]. Therefore, the formations of hydrogen, methane, water, carbon monoxide and carbon dioxide are mainly due to the scissions of the methylene groups at high temperatures [31].

Figure 4a,c shows that the TG-MS results of both phenolic resins are completely consistent with the research results of Trick et al. [22], and the curves of hydrogen, carbon dioxide, carbon monoxide and methane are basically consistent under argon atmosphere. The introduction of POSS improves the instability of methylene groups, resulting in improvements in the emission temperatures of the above mentioned volatiles.

Due to the presence of oxygen, the methylene groups are more easily broken under air than under argon. Oxygen can directly react with hydrogen and carbon in methylene groups to produce oxygen-containing volatiles such as water, carbon monoxide and carbon dioxide [31,32,36]. Thermal oxidation causes the oxygen-containing volatiles to be emitted at lower temperatures under air. By comparing the mass spectra of phenolic resin under air and under argon, it can be seen that the release temperatures of the oxygen-containing volatiles in Figure 4b are significantly lower than those of the oxygen-containing volatiles in Figure 4a.

As shown in Figure 4c,d, the modification effect of POSS on the thermal stability is more obvious under air than that under argon. Under argon, the effect of POSS on improving the thermal stability of phenolic resin is to increase the structural stability of the main chain. However, under air, POSS can absorb oxygen at high temperatures to form solid products such as silica, which greatly weaken the thermal oxidation reactions between oxygen and methylene. Owing to the oxidation–reduction of POSS, the maximum peak of carbon monoxide appears at about 570 °C under air, which is similar to the maximum released temperature of carbon monoxide under argon.

### 3.4. Degradation Residue Analysis

To confirm the phase compositions of the degradation residues, both phenolic resins and their solid products after pyrolysis are investigated under different atmospheres by an XRD analysis. The corresponding X-ray diffraction patterns are shown in Figure 5.

At room temperature, the XRD patterns of both neat phenolic resin and POSS modified phenolic resin appear one broad peak at a 2θ angle of 18.69°. The broad peak corresponds to the crystalline polymer on the carbon chain after curing [37]. After thermal degradation under argon, two broad peaks appear at a 2θ angle of 23.38° and 44.10° [38,39], corresponding to the (002) and (100) lattice planes for char (PDF26-1079). This reflects the fact that not only neat phenolic resin but also POSS modified phenolic resin has been charred by pyrolysis at high temperatures under argon.

After thermal degradation under air, the degradation residues of neat phenolic resin still show the characteristic peaks of char in their XRD pattern. However, as shown in the XRD pattern of POSS modified phenolic resin, besides the characteristic peaks of char, the characteristic peaks of SiO_2_ (PDF 65-0466) are observed in the degradation residues. The presence of SiO_2_ is mainly attributed to the reaction between POSS and oxygen at high temperatures. Hence, POSS has an oxidation–reduction effect on the thermal degradation process of phenolic resin, resulting in the improvement of its thermal stability under air atmosphere.

## 4. Conclusions

A POSS modified phenolic resin is synthesized as a matrix of thermal protection material to enhance its thermal stability. The thermal stability and the thermal degradation products of POSS modified phenolic resin are investigated by TGA, TG-MS and XRD. Neat phenolic resin is also fabricated to compare the structure and performance with POSS modified phenolic resin over the whole process of study. The major conclusions are presented below.

The introduction of POSS makes phenolic resin show better thermal stability under air. The residual char yield of POSS modified phenolic resin at 800 °C is 14.73% higher than that of phenolic resin. However, under argon atmosphere, the residual char yield of POSS modified phenolic resin at 800 °C is only 6.95% higher than that of phenolic resin.Under argon, phenolic resin releases the degradation volatiles at around 580–610 °C, while POSS modified phenolic resin does not reach the maximum speed until 630 °C. The corresponding volatiles evolutions under air are more complex than those under argon. Under air, the volatiles release rate of phenolic resin is the fastest at around 480–520 °C, while volatiles produced by the thermal degradation of POSS modified phenolic resin reach the maximum peaks at 540–610 °C.The degradation processes of both phenolic resins are carbonization processes, and the solid phase in the residual products is mainly composed of char. In addition to the char, crystalline SiO_2_ is detected in the residue of POSS modified phenolic resin after degradation under air. The existence of SiO_2_ is mainly attributed to the reaction of POSS with oxygen at high temperatures, and the existence of SiO_2_ can effectively improve the high temperature oxidation resistance of phenolic resin.

## Figures and Tables

**Figure 1 polymers-13-01182-f001:**
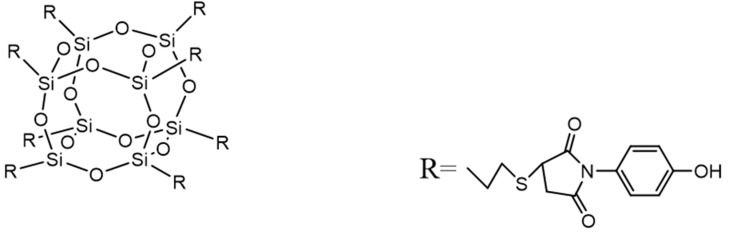
Octaphenol-polyhedral oligomeric silsesquioxane (POSS-8Phenol).

**Figure 2 polymers-13-01182-f002:**
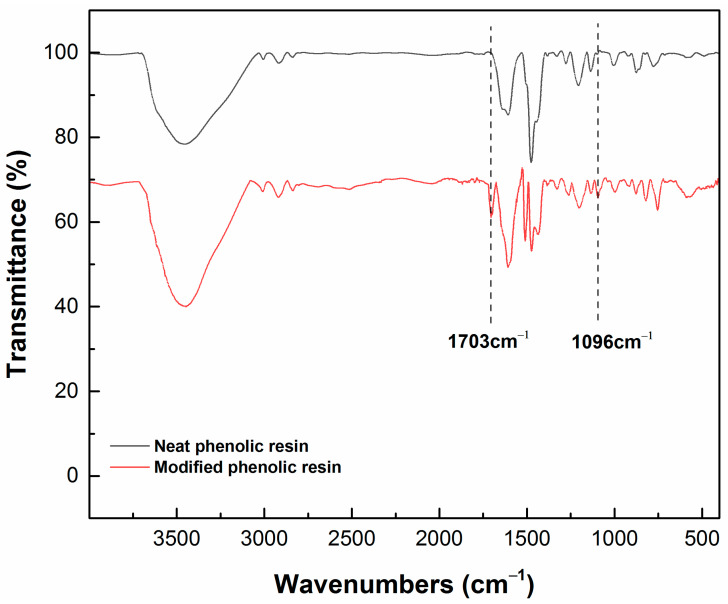
IR spectra of phenolic resin before and after the modification.

**Figure 3 polymers-13-01182-f003:**
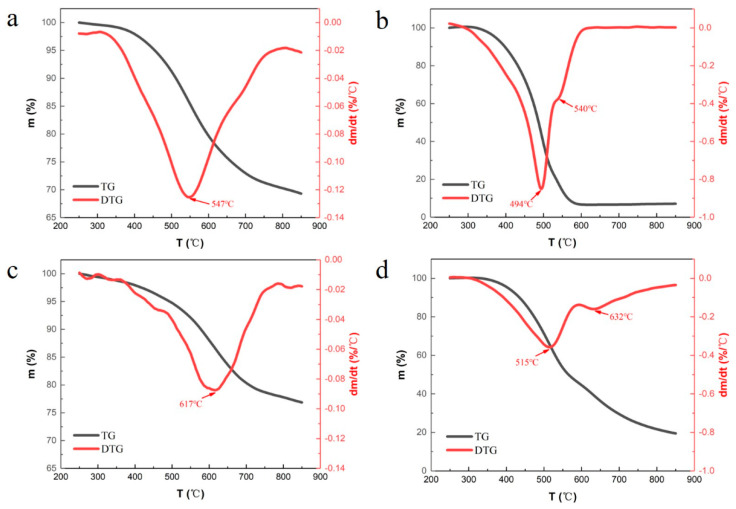
Thermogravimetric curves of both phenolic resins under different conditions. (**a**) Neat phenolic resin under argon; (**b**) Neat phenolic resin under air; (**c**) POSS modified phenolic resin under argon; (**d**) POSS modified phenolic resinunder air.

**Figure 4 polymers-13-01182-f004:**
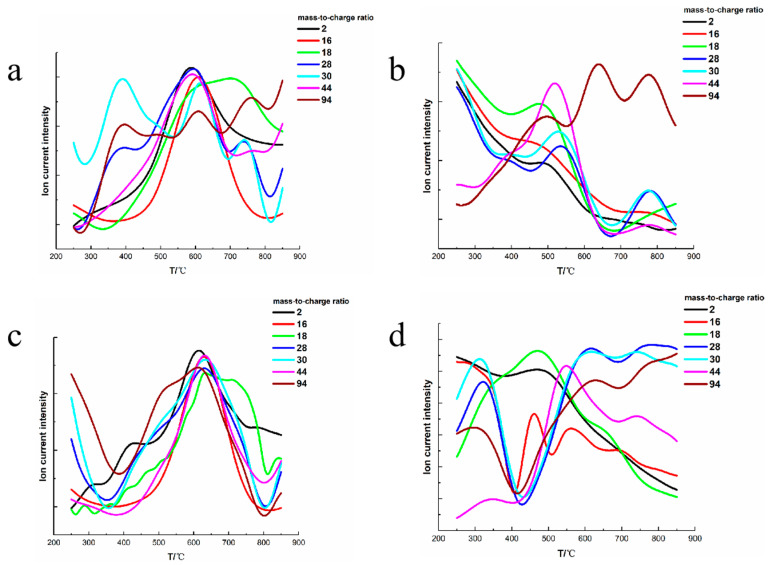
Mass spectra of both phenolic resins under different atmospheres. (**a**) Neat phenolic resin under argon; (**b**) Neat phenolic resin under air; (**c**) POSS modified phenolic resin under argon; (**d**) POSS modified phenolic resin under air.

**Figure 5 polymers-13-01182-f005:**
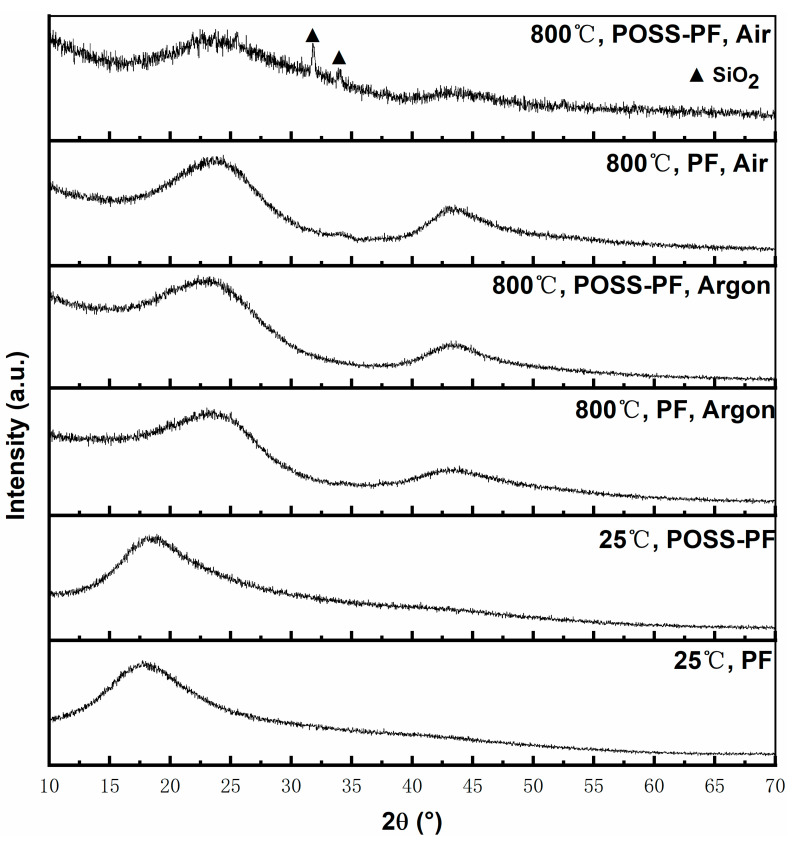
XRD spectra of neat phenolic resin and POSS modified phenolic resin under different conditions.

## Data Availability

The data presented in this study are available on request from the corresponding author.

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
