# Peer review of "Synthesis and Thermal Degradation Study of Polyhedral Oligomeric Silsesquioxane (POSS) Modified Phenolic Resin"

_polymers, 2021, doi:10.3390/polym13081182_

Round 1

Reviewer 1 Report

This manuscript presented a study about the synthesis and thermal degradation of POSS modified phenolic resin. The work has some potential. However, several points listed below need to be improved.

Abstract: I suggest add more numerical results to the abstract.

Introduction – Line 30: what c/c composite materials means?

Section 2.2 – Line 91: what the MPT/methanol solution ratio used?

Instruments section:

  1. a) the number of this section must be changed to 2.3. In addition, I suggest used Sample characterization instead of “Instruments”.
  2. b) What was the number of scans used in FTIR analysis.
  3. c) What are the conditions (mass, heating rate, temperature,…) used in TGA?
  4. d) What are the parameters used in XRD analysis?

Section 3.1: the FTIR results must be better discussed.

Section 3.3 – Lines 166-200: I suggest add this part to the methodology section.

Figure 5: please describe in the figure what the red, black, green and blue lines represent.

Table 1: I suggest use only two digits after comma for Tp values.

Lines 221-226: this paragraph is unclear. Please rewrite.

Table 1, Table 2 and Figure 6: the results presented in these tables and figure must be deeply discussed. Better discuss the effect of the atmosphere on the sample thermal degradation.

Section 3.4: the conditions and equipment used in mass spectrometry must be described in section “Instruments”.

Section 3.5: the authors must improve the discussion of XRD results.  

Reviewer 2 Report

Dear Authros,

The manuscript is interesting, but the lack of deeper discussion concerns me.

In the last sentence of the introduction, it must be rewritten so that the reader does not have a conclusion of the work before its discussion.

Standardize the units and the number of decimal places in the amount of reagents used in the methodology. Please check spaces between words.

The IR spectra should be better discussed, especially in the 1700-1000 cm-1 region.

TG analysis, XRD, cracking mechanism data must be referenced according to the literature in order to enrich the discussion. In addition, the characterizations should be presented by completing the information.

The manuscript has only 17, which 10 are in the introduction. References should be reviewed and standardized according to the guidelines of the journal.

Author Response

Point 1: The manuscript is interesting, but the lack of deeper discussion concerns me.

Response 1: As suggested by the reviewer, the original manuscript lacked the deeper discussion, and therefore the discussion has been rewritten.

In the revised manuscript, the discussion part is divided into five parts: Synthesis of modified phenolic resin, Thermogravimetric analysis, Degradation kinetics, Degradation volatiles analysis, Degradation residue analysis.

In Synthesis of modified phenolic resin, the characteristic peaks before and after modification in the FTIR spectra have been deeply analyzed and discussed again. On this basis, we point out the respective attributions of the characteristic peaks.

In Thermogravimetric analysis, the thermal degradation behaviors of both phenolic resins under different atmospheres were re-analyzed. The differences of thermal degradation behavior and degradation residue at high temperature were discussed emphatically.

In Degradation kinetics, the results presented in Table 1, Table 2 and Figure 6 have been deeply discussed. We put the emphasis on the effect of the atmosphere on the thermal degradation process.

In Degradation volatiles analysis, the thermal degradation mechanism is further studied from the perspective of the gas products. The effects of atmosphere on the temperature of gas products are summarized. The oxidation of phenolic resin by oxygen at high temperature is analyzed, and the oxidation–reduction of modified phenolic resin by POSS at high temperature is also analyzed,.

In Degradation residue analysis, we confirm the thermal degradation mechanism from the perspective of the solid residue. The results of XRD analysis are discussed again. The characteristic peaks of X-ray diffraction are marked and matched to PDF card.

Point 2: In the last sentence of the introduction, it must be rewritten so that the reader does not have a conclusion of the work before its discussion.

Response 2: As suggested by the reviewer, the last paragraph of the introduction has been rewritten. The last paragraph of the introduction is aimed to the research ideas and research methods on this paper.

In line 72-83 of the revised manuscript, the revised paragraph is as follows:

“In this article, we used the hydrolysis and condensation of trimethoxysilane to prepare POSS-8SH. Then the POSS-8Phenol, which contains octagonal phenol, was synthesized through a mild and efficient Michael addition reaction between sulfhydryl and double-bonded. After this, POSS-8Phenol, as a special phenol, was introduced into the molecular structure of thermoset phenolic resin. The chemical structure of POSS modified phenolic resin was studied by IR. The kinetics of degradation reaction of POSS modified phenolic resin was studied by TGA. By means of Gaussian-peak analysis of the relationship between weightlessness rate and temperature, the kinetic parameters of thermal degradation were calculated, and the kinetic equations of thermal degradation were established under argon and air. Finally, the volatiles and residues of POSS modified phenolic resin were characterized by TG-MS and XRD in order to further understand the degradation mechanism. Neat phenolic resin was fabricated into subsample and investigated with the same analysis process.”

Point 3: Standardize the units and the number of decimal places in the amount of reagents used in the methodology. Please check spaces between words.

Response 3: As suggested by the reviewer, we have standardized the units and the number of decimal places. Numerical results are unified to two decimal places. There is a space between the number and the unit, except for the percent sign.

Point 4: The IR spectra should be better discussed, especially in the 1700-1000 cm-1 region.

Response 4: As suggested by the reviewer, we analyzed the same characteristic peaks before and after modification in the FTIR spectra, and pointed out their respective attributions, especially range from 1000 cm-1 to 1700 cm-1.

In line 174-185 of the revised manuscript, the supplementary text is as follows: “The IR spectra of neat phenolic resin and modified phenolic resin are recorded to confirm the chemical structure of POSS in modified phenolic. The absorption peak at 1703 cm-1 assigned to C=O in maleimide group of POSS-8Phenol. And the characteristic of Si-O-Si stretching is at 1096cm-1, which is consistent with the literature[28].Other infrared absorption peaks are basically the same before and after modification, the distribution of characteristic peaks is as follows: the characteristic peaks of O-H of the phenol group and -CH2OH are at around 3500 cm-1 (broad peak); the C-H stretching vibration absorption peaks in -CH2- range from 2800 cm-1 to 2950 cm-1; the stretching vibration peaks of the C=C double bond in the benzene ring appear at 1610 cm-1 and 1510 cm-1; the characteristic absorption peaks of C-O in phenolic group is at 1270 cm-1; the absorption peaks of aliphatic C-O stretching vibration are at 1130 cm-1 and 1005 cm-1; the ortho- and para-substituted peaks of the benzene ring are at 760 cm-1 and 870 cm-1. The above results have confirmed that POSS-8Phenol has been incorporated into the structure of phenolic resin.”

Point 5: TG analysis, XRD, cracking mechanism data must be referenced according to the literature in order to enrich the discussion. In addition, the characterizations should be presented by completing the information.

Response 5: As suggested by the reviewer, TG analysis, XRD and cracking mechanism data have been referenced to the literatures .

In TG analysis and cracking mechanism data, we refer to Trick KA's research result. The reference is “Trick KA, Saliba TE. Mechanisms of the pyrolysis of phenolic resin in a carbon/phenolic composite. Carbon1995, 33(11), 1509-15.”

The main contents are the three stages of thermal degradation for phenolic resin, and the thermal degradation mechanism of each stage. These results are compared with our results.

In XRD analysis and cracking mechanism data, we refer to Liu's research result. The reference is “Liu, Y.; Jing, X. Pyrolysis and structure of hyperbranched polyborate modified phenolic resins. Carbon 2007, 45, 1965-1971.”

We compare our XRD patterns with those of Liu, and the emphasis is on the comparison of the 2θ angle of char. The char is formed by the pyrolysis of phenolic resin. Moreover, we focus on the XRD pattern of the POSS modified phenolic resin. After thermal degradation under argon, the diffraction characteristic peaks of the solid residues are matched with PDF cards one by one. A new phase SiO2 (PDF 65-0466) has been found, which further prove the oxidation–reduction effect of POSS during the thermal degradation process of phenolic resin.

Point 6: The manuscript has only 17, which 10 are in the introduction. References should be reviewed and standardized according to the guidelines of the journal.

Response 6: As suggested by the reviewer, we have carefully revised the paper from the beginning to the end. The introduction part only takes 2 pages, and the discussion part takes up 7 pages. We have added appropriate references to support this paper, and the number of references is 32. In the revised manuscript, references have been standardized according to the MDPI reference list style by Note express software.

Round 2

Reviewer 1 Report

After corrections the manuscript reads well. I suggest publication.

Author Response

Thank you very much for your help during our submission, thank you for your review, your comments. The manuscript has been revised again. Please review it.

Reviewer 2 Report

Dear Authors,

The quality of the manuscript has improved, however my main concern is the results that are rarely discussed with the literature. I understand that the authors discuss the results, but comparison with the data in the literature is essential. I still notice that there was an increase in references, but they are almost all in the introduction instead of in the results. The abstract is still confused.
